# Impact of Obstructive Sleep Apnea Syndrome on Ventricular Remodeling after Acute Myocardial Infarction: A Proof-of-Concept Study

**DOI:** 10.3390/jcm11216341

**Published:** 2022-10-27

**Authors:** François Bughin, Hélène Kovacsik, Isabelle Jaussent, Kamila Solecki, Sylvain Aguilhon, Juliette Vanoverschelde, Hamid Zarqane, Jacques Mercier, Fares Gouzi, François Roubille, Yves Dauvilliers

**Affiliations:** 1PhyMedExp, University of Montpellier, INSERM, CNRS, CHU, 34090 Montpellier, France; 2Pneumology Department, Clinique du Millénaire, 34000 Montpellier, France; 3Department of Interventional and Cardiovascular Imaging, CHU, 34090 Montpellier, France; 4Institute for Neurosciences of Montpellier INM, University of Montpellier, INSERM, 34000 Montpellier, France; 5Cardiology Department, Clinique Beausoleil, 34070 Montpellier, France; 6Cardiology Department, INI-CRT, CHU de Montpellier, PhyMedExp, Université de Montpellier, INSERM, CNRS, 34090 Montpellier, France; 7Unité du Sommeil, Centre National de Référence pour la Narcolepsie, CHU Montpellier, Hôpital Gui-de-Chauliac, Service de Neurologie, 34090 Montpellier, France

**Keywords:** obstructive sleep apnea syndrome, acute myocardial infarction, ventricular remodeling

## Abstract

Background: Obstructive sleep apnea syndrome (OSA) is common in patients with acute myocardial infarction (AMI). Whether OSA impacts on the ventricular remodeling post-AMI remains unclear. We compared cardiac ventricular remodeling in patients assessed by cardiac magnetic resonance (CMR) imaging at baseline and six months after AMI based on the presence and severity of OSA. Methods: This prospective study included 47 patients with moderate to severe AMI. They all underwent CMR at inclusion and at six months after an AMI, and a polysomnography was performed three weeks after AMI. Left and right ventricular remodeling parameters were compared between patients based on the AHI, AHI in REM and NREM sleep, oxygen desaturation index, and daytime sleepiness. Results: Of the 47 patients, 49% had moderate or severe OSA with an AHI ≥ 15/h. No differences were observed between these patients and those with an AHI < 15/h for left ventricular end-diastolic and end-systolic volumes at six months. No association was found for left and right ventricular remodeling parameters at six months or for the difference between baseline and six months with polysomnographic parameters of OSA severity, nor with daytime sleepiness. Conclusions: Although with a limited sample size, our proof-of-concept study does not report an association between OSA and ventricular remodeling in patients with AMI. These results highlight the complexity of the relationships between OSA and post-AMI morbi-mortality.

## 1. Introduction

Ischemic heart disease remains the first cause of death in adults in most developed countries [1]. Even though death rates following acute myocardial infarction (AMI) have fallen substantially in the past decades with therapeutic advances [2], survivors of AMI remain at risk of cardiovascular events including death, recurrent myocardial infarction [3], heart failure [4], arrhythmias, angina, and stroke. These adverse events may be underpinned by a ventricular remodeling, defined by the structural and functional changes of the myocardium occurring after the cardiac injury [5]. Disease-related ventricular remodeling is a complex process involving cardiac myocyte growth and death, vascular rarefaction, fibrosis, inflammation, and electrophysiological remodeling [5]. Consistently, this ventricular remodeling appeared as a key factor of the patient’s prognosis after AMI [6]. Identifying the risk factors of post-AMI ventricular remodeling is thus a critical step in research aiming to improve the patient’s prognosis in AMI. Obstructive sleep apnea (OSA) is a sleep-related breathing disorder that increases the risk of cardiovascular diseases [7,8] which is very common in more than half of patients with AMI [9,10], and may be a risk factor of post-AMI ventricular remodeling.

A pejorative role of OSA on post-AMI ventricular remodeling has been hypothesized via its negative impact on clinical outcomes in patients with AMI. Indeed, studies have shown that the presence of OSA may worsen the prognosis of patients admitted for AMI both in the short [11] and long term [12,13]. Each episode of upper airway obstruction and respiratory efforts during sleep causes arousals and sleep fragmentation, and leads to an increase of sympathetic activity, systemic inflammation, and oxidative stress, and changes in left ventricular transmural pressure [14]. All these factors have been identified as deleterious for ventricular remodeling [15]. However, these results have been challenged by studies showing the cardio-protective role of sleep apnea during AMI, via an ischemic preconditioning [16]. One study using cardiac magnetic resonance (CMR) imaging showed that sleep disordered breathing was associated with less salvaged myocardium and a smaller reduction in infarct size within three months after AMI; however this study was never replicated [17]. Altogether, the impact of OSA on the ventricular remodeling post-AMI remains to be further studied.

The aim of this study was to analyze the association between OSA and the cardiac ventricular remodeling in post-AMI patients, comparing changes in cardiac morphology using CMR imaging at baseline and six months after AMI based on the presence and severity of OSA characterized by the apnea hypopnea index (AHI), AHI in rapid eye movement (REM) and nonrapid eye movement (NREM) sleep, oxygen desaturation and excessive daytime sleepiness (EDS).

## 2. Materials and Methods

### 2.1. Study Design

This prospective study included patients admitted for AMI to the cardiological Intensive Care Unit of the University Hospital of Montpellier (France). CMR was scheduled 5 ± 2 days after the percutaneous coronary intervention and at six months. Polysomnography was carried out three weeks after the percutaneous coronary intervention. A clinical standardized evaluation including sleep parameters was performed for all patients. Blood samples were carried out on the day of admission for AMI. The study was conducted in accordance with the CONSORT ethical guidelines and was approved by the local ethics committee (CPP Sud-Mediterrannee IV, N° ID-RCB: 2015-A00079-40). All participants gave their informed written consent (ClinicalTrials.gov identifier NCT02439294).

### 2.2. Patient and Public Involvement

No patients or members of the public were involved in the design, conduct, reporting, or dissemination plans of our research

### 2.3. Study Population

The inclusion criteria were adults aged between 18 and 89 years, hospitalized at the Montpellier University Hospital in Intensive Care Units for AMI (detection of rise and fall of cardiac biomarkers with at least symptoms of ischaemia, ECG changes, or imaging abnormalities [18]) confirmed by coronary angiography, and moderate to severe AMI defined by a score of delayed enhancement CMR ≥ 5 segments of 17. Key exclusion criteria were a mild infarction (rate of delayed enhancement CMR ≤ 4 segments of 17), patient treated with continuous positive airway pressure (CPAP) prior AMI, sleep apnea syndrome whose central part was predominant (>50%), early implantation of a cardiac device, or other contraindications for CMR.

### 2.4. Outcomes

The primary endpoint was the difference in left ventricular remodeling assessed by the changes of end-diastolic and end-systolic left ventricular volumes in CMR six months after AMI between patients with and without OSA.

Secondary outcomes were the association between CMR-based left and right ventricular remodeling parameters and clinical, biological, and polysomnographic parameters.

### 2.5. Measures

#### 2.5.1. Cardiac Magnetic Resonance Imaging

All CMR studies were performed on a 1.5 T AREA^®^ system (Siemens, Erlangen, Germany) using vectocardiogram monitoring and a phased-array cardiac receiver coil. Localizers and left ventricular functional assessment were performed using steady-state free-processing sequences in the three axes of the heart. Multiplanar T2 STIR acquisition was performed to study oedema and areas at risk. In the short-axis orientation, the left ventricle was completely encompassed by contiguous slices with coverage of the entire left ventricular (LV) and right ventricular (RV), reference method for calculating both right and left ventricular, end-systolic and end-diastolic volumes, myocardial mass, and left ventricular ejection fraction (LVEF). Calculation of end-systolic volume (ESV), end-diastolic volume (EDV), and ejection fraction (EF) in multiphase series was performed using CVI42 software (Circle Cardiovascular Imaging, Calgary, AB, Canada). Based on recent studies [19,20], a cardiac remodeling could be defined as an increase of left ventricular end-diastolic volume (LVEDV) ≥ 12% after six months.

Inversion recovery and PSIR sequences were acquired 10 min after administration of 0.2 mmol/kg gadolinium-based contrast agent (Dotarem^®^, Guerbet, Roissy CdG, England, France) to assess the extent of the infarction (number of segments), the transmurality and the presence of no reflow (i.e., microvascular dysfunction). The viability criteria studied were transmural enhancement and the presence of no-reflow. All the examinations were interpreted by experienced radiologists.

#### 2.5.2. Polysomnography

All patients underwent a polysomnography in the sleep laboratory in Montpellier University Hospital. PSG included EEG leads, electrooculogram, electromyography of the chin and tibialis anterior muscle, electrocardiogram, nasal cannula and pressure transducer, mouth thermistor, chest and abdominal bands, and pulse oximeter. Obstructive and central apneas were defined as an airflow cessation (>90%) for more than 10 s associated with or without thoracoabdominal movements, and hypopnea as airflow reduction of more than 30% associated with a drop in O_2_ saturation of more than 3% or a microarousal. Sleep stage, microarousals, periodic limb movements (PLM), and respiratory events were scored manually according to standard criteria [21]. Mild, moderate, and severe OSA were respectively defined by an AHI above 5, between 15 and 29.9, and greater or equal to 30 events per hour of sleep. Apnea central syndrome, defined by an AHI ≥ 5/h with more than 50% of respiratory events of central origin, was a noninclusion criterion.

#### 2.5.3. Blood Tests

Blood samples were taken on the day of admission for AMI to measure glycemia, high-sensitivity C-reactive protein hs-CRP, HbA1c, cholesterol (HDL and LDL), triglycerides, hs-troponin T, CPK, NT-proBNP, creatinine, and ionogram.

#### 2.5.4. Self-Reported Questionnaires

Epworth sleepiness scale (ESS) [22]: This questionnaire assesses EDS through situations frequently found in everyday life during which the propensity to fall asleep is measured on a scale from 0 to 3. The score is noted from 0 to 24. A score of >10 defines EDS.

Insomnia Severity Index (ISI) [23]: This questionnaire estimates the severity of insomnia symptoms over the past month. The severity of insomnia symptoms is a function of the total score; not clinically significant between 0 and 7, mild between 8 and 14, moderate between 15 and 21, and severe above 22.

EQ5D [24]: The EQ-5D is a preference-based health survey assessing five health dimensions (with three levels of problems) and an overall health visual analog scale (EQ-VAS) which assesses the respondent’s self-rated health status on a graduated (0–100) scale, with higher scores for higher health-related quality of life.

Beck depression inventory [25]: This questionnaire describes the patient’s mood over the two weeks preceding the visit. A score between 10 and 18 indicates minor, 19 and 29 indicates moderate, and 30 and above indicates severe depressive symptoms.

### 2.6. Power Calculation

In the absence of studies carried out with a comparable methodology at the time of drafting the protocol (in 2013), statistical power was based on the estimated distribution of left ventricular remodeling at six months in a group of patients with AMI with a mean of 84 mL and a standard deviation of 29 [26]. Therefore, 72 patients (24 patients without OSA and 48 patients with OSA) provided a power of 0.80 for detecting a between-group difference in the left ventricular remodeling of 25% at six months (mean of the left ventricular remodeling at six months of 84 mL for patients without OSA and 63 mL for patients with OSA) using a two-sided ratio 1–2 with a significance level of 0.05. The rate of lost to follow-up being estimated at 20%, a total of 87 patients was therefore necessary for this study.

### 2.7. Statistical Analysis

The characteristics of the study population were described using number and percentage for categorical variables and median (minimum value—maximum value) or IQR for continuous variables due to their distributions being mostly skewed according to Shapiro–Wilk test. Nonparametric statistical tests were used due to the small sample size (less than 30 in each group). The Mann–Whitney test was performed to compare continuous variables of two groups of the AHI and the Kruskal–Wallis test to compare the four groups (none, mild, moderate, and severe OSA). Chi-square or Fisher’s exact tests were used to compare categorical variables between the groups of the AHI. When comparisons were statistically significant between groups, two-by-two comparisons were carried-out, using a correction for multiple comparisons with the Bonferroni method. Changes in each outcome between baseline and a six-month follow-up were compared between the groups using a Mann–Whitney U test. For the between-group differences, 95% confidence intervals (CI) were constructed. To compare differences between baseline and six-month follow-ups within groups, Wilcoxon signed-rank tests were used. Spearman’s rank order correlations were used to determine associations between continuous variables. Statistical significance was set at *p* < 0.05. Statistical analyses were performed with SAS version 9.4 (SAS Institute, Cary, NC, USA).

## 3. Results

Between May 2015 and January 2020, 65 patients who underwent percutaneous coronary intervention to treat AMI were eligible for the prospective study. Among them, 18 patients were excluded due to the absence of CMR at six months (*n* = 6) or baseline polysomnography recording (*n* = 5), a SAS with predominant central part (*n* = 5), a cancer (*n* = 1), or a consent withdrawal (*n* = 1) (Figure 1).

The final sample included 7 patients, 85% were male with a median age of 57.4 years (min = 33.6, max = 74.4) and a median body mass index (BMI) of 25.2 kg/m^2^ (min = 18.5, max = 33.1). The infarct-related artery was left anterior descending, circumflex, and right coronary artery for, respectively, 38 (81%), 2 (4%), and 4 (9%) of patients. Three patients (6%) had two arteries responsible for the infarction. All patients had an ST-elevation myocardial infarction (STEMI).

All patients had CMR at baseline (median = 6 days, interquartile range (IQR) = 4 days) after the AMI and underwent a polysomnography in a median delay of 28 days (IQR = 12) after AMI. The second CMR was performed with a median delay of 6.3 months (IQR = 1.1).

The severity of AMI was confirmed by the large proportion of patients with nonviability criteria on the initial CMR: 72% patients had no reflow and 96% patients had transmural enhancement ≥ 75%. Among the overall population, respectively, 27%, 15% and 49% had EDS (ESS score > 10), insomnia symptoms (ISI > 14), and a low quality of life (EQ-5D VAS < 60).

Patients were initially divided into four groups: no OSA (AHI < 5/h, *n* = 7), mild (5 ≤ AHI < 15/h, *n* = 17), moderate (15 ≤ AHI < 30, *n* = 12), or severe OSA (AHI ≥ 30/h, *n* = 11) (Table 1 and Table 2). The 11 patients with severe OSA were offered a CPAP, but only one patient used CPAP more than 2 h per night at six months. Given the small sample, we then categorized the patients into two groups based on the AHI (AHI < 15/h and AHI ≥ 15/h). No differences were found between patients with an AHI ≥ 15/h (*n* = 23) and those with AHI < 15/h (*n* = 24) for demographic, clinical, and biological characteristics of the patients except for a higher level of hs-CRP, a better quality of life as well as a trend for more hypercholesterolemia in patients with an AHI ≥ 15/h (Table 1). Moreover, these patients had higher hypopnea, obstructive and central sleep apnea indexes, AHI in REM and NREM sleep, and higher oxygen desaturation and microarousal indexes (Table 2).

### Outcome Measures at Six Months

No significant differences were observed on CMR imaging at six months between patients with and without OSA defined by AHI ≥ 5/h or ≥15/h for left ventricular end-diastolic volume (LVEDV) and left ventricular end-systolic volume (LVESV) (Figure 2, Table 3). We found no differences in other LV and RV remodeling parameters at six months and for the difference between baseline and six months between patients with and without OSA (Figure 2, Table 3).

Based on their levels of AHI_REM_, AHI_NREM_, oxygen desaturation index (ODI), mean nocturnal peripheral oxygen saturation (SpO_2_), EDS, and EDS associated with an AHI ≥ 5/h or ≥15/h, no significant differences were found between patient groups on the changes in LV and RV parameters at six months (Figure 3). No significant associations were found between ventricular remodeling parameters and clinical parameters (e.g., insomnia symptoms, depressive symptoms, or quality of life), sleep architecture (e.g., sleep efficiency, total sleep duration, N3 and REM sleep percentages, and microarousal index) and biology (inflammation, lipids, and glycemia).

Among the whole sample, 19 (40%) patients had an increase of LVEDV ≥ 12% between baseline and six months. These patients had a lower rate of N3 sleep (*p* = 0.03) compared to patients without ventricular remodeling. However, based on this threshold of 12%, no differences in OSA severity (AHI, AHI_REM_, AHI_NREM_, O_2_ desaturation, and EDS) were found between patients who had or had not remodeled their left ventricle.

Unfortunately, the final sample included 47 patients instead of the 72 patients (24 patients without OSA and 48 patients with OSA) originally planned for detecting significant between-group differences in the left ventricular remodeling at six months. The low sample size can be explained by the absence of routine CMR in patients diagnosed with AMI, the inclusion of severe AMI patients (baseline CMR delayed enhancement score ≥ 5 segments), the inability to recruit patients during the COVID-19 pandemic, and by the difficulties for patients to return for a CMR reevaluation when living far away from our tertiary center. However, achieving the expected sample size would have shown no significant effect of OSA severity on the post-AMI ventricular remodeling. A post hoc power analysis showed that 1032 patients would be required to reach a power of 0.8 with a significant level of 0.05 considering a mean LVEDV = 189 mL for the seven patients with an IAH < 5/h, a mean LVEDV = 180.38 mL for the 40 patients with an IAH ≥ 5/h, and a standard deviation at 51.54 for the 47 patients. Such recruitment is beyond the scope of this study.

## 4. Discussion

Our proof-of-concept study with a limited sample size found no association between the markers of OSA severity (AHI, AHI_REM_, AHI_NREM_, ODI, and EDS) and ventricular remodeling assessed by CMR at six months or between changes at six months from baseline after AMI.

We found a high frequency of OSA in our sample assessed via polysomnography within one month of AMI, with 85% of patients having an AHI ≥ 5/h and 49% having moderate to severe OSA. These results were in agreement with the large prevalence of OSA in this condition [9,10]. Moreover, 40% of AMI patients displayed a ventricular remodeling characterized by an increase of ≥12% of LVEDV [19]. This frequency was higher compared to other studies [19], probably due to the inclusion of severe AMI patients (CMR delayed enhancement score ≥ 5/17 segments at baseline.

Our study did not reveal differences in the left or right ventricle remodeling between severity groups of OSA patients [17]. A single CMR study included 56 patients and demonstrated that sleep-disordered breathing was associated with less myocardial salvage following AMI [17]. However, in this study, the follow-up was limited to three months, and patients were affected with either obstructive or central sleep apnea syndrome. A second study was performed but to re-evaluate the CRM images with an alternative method [27], the sphericity index, with images obtained from the previous study. Both studies did not assess the association between CRM images and clinical parameters such as EDS and quality of life, the impact of apnea or hypopnea in REM and NREM sleep independently, O_2_ desaturation, and the biological parameters (e.g., low-grade inflammation). Moreover, this study performed the polysomnography earlier after AMI, at three to five days versus almost one month in our study, which may also explain some differences in the frequency of central vs. obstructive apnea or hypopnea events, the former often evolving favorably over time [28,29]. Patients with predominant central sleep apnea syndrome were excluded from our study (AHI ≥ 5/h with more than 50% of respiratory events of central origin); however, some central apnea events were noted in our patients, being higher in those with an AHI ≥ 15/h. An old study that included 86 AMI patients with PSG assessment at 2–3 weeks and left ventriculograms at baseline and three weeks reported that OSA (defined by AHI ≥ 15/h) may inhibit the recovery of LV, with a correlation between AHI and delta of LVEF [30].

We found here a lack of effect of OSA on the ventricular remodeling assessed at six months after AMI. Given the weight of the ventricular remodeling on the cardiovascular morbidity post-AMI [20], these results could translate into an absence of increased risk of major cardiovascular events post-AMI in OSA patients. One prospective study involving 112 patients revealed that in the absence of EDS, AMI patients with moderate to severe OSA (AHI ≥ 15/h) were not associated with an increased probability of major cardiovascular events at 36 months [31]. In our study, EDS was not associated with ventricular remodeling in patients with or without moderate to severe OSA. Additionally, a randomized controlled trial in AMI patients reported that treating OSA with CPAP did not reduce the frequency of major cardiovascular events [32].

In addition to the post-AMI condition, sleep cohort studies and meta-analyses reported associations between OSA and long-term ventricular remodeling [33]. In the Wisconsin sleep cohort study, OSA was associated with a slight decrease in LV systolic function when reevaluated 18.0 ± 3.7 years after the OSA diagnosis. Mean nocturnal O_2_ desaturation and percent sleep time with O_2_ saturation < 90% were independent predictors of LV mass and LV wall thickness [34]. In a large, community-based cohort study using CMR imaging, Javaheri et al. showed an increased LV mass with increasing AHI categories, but no association between OSA severity and LVEF 10 years after the diagnosis. In this study, O_2_ saturation was associated with LV mass to volume ratio [35]. These two cohort studies highlighted the impact of intermittent hypoxia on the ventricular hypertrophy. Yet, the duration of intermittent hypoxia exposure has not been assessed in post-AMI studies specifically. However, the intermittent hypoxia related to OSA could also have cardioprotective effects in AMI patients. One study demonstrated that patients with OSA have less severe cardiac injury during an acute nonfatal MI when compared to patients without OSA, suggesting a cardioprotective role of sleep apnea during AMI via ischemic preconditioning [16]. In vivo intermittent hypoxia conditioning has shown to improve the postischemic cardiac function in animal models [36]. Ventricular remodeling may also be driven by other factors such as inflammation [37]. Our study found association between AHI and low-grade inflammation with increased hs-CRP levels, but not with the ventricular remodeling. Accordingly, although colchicine, a pleiotropic anti-inflammatory drug, has been shown to be efficient in patients with coronary artery disease [38,39], it failed to reduce infarct size and LV remodeling [40] after a large MI.

### Strengths and Limitations

Our study had several strengths. CMR is considered the gold standard method for evaluating ventricular remodeling with low inter-reader variability and high reproducibility [41]. Different definitions of adverse LV remodeling exist, however we used LVEDV and LVESV as primary outcomes based on their good predictive values for long-term clinical outcomes [15,19]. Unlike the previous unreplicated study [17], we assessed all parameters related to OSA severity including clinical, polysomnographic, and biological parameters. Despite this comprehensive assessment, we found no association between OSA severity and ventricular remodeling.

Our study has some limitations. The main limitation is the sample size with a risk of type II error, a low recruitment of 47 patients (65% of the planned sample size), 10% lost during follow-up, a wide range of BMIs in the patients, an underrepresentation of females, and a monocentric study design. Only one patient (9%) with severe OSAS used CPAP for more than two hours per night at six months. This finding was in agreement with the ISAAC study [32] that highlighted the limited impact of CPAP in this specific population. Finally, our study addressed the cardiac remodeling of structure and functional parameters as a function of OSA severity, with a limited population of subjects, a monocentric study design, the inclusion of severe AMI patients, and polysomnography performed three weeks after AMI. Cardiac remodeling is influenced by multiple factors such as coronary disease and extension of ischemia; thus our negative results could relate to cohort selection requiring ≥5 segments with late enhancement at CMR, which may render less relevant the effect of OSA on remodeling. Finally, our results may be not generalizable, they remain limited to six months and therefore not addressed in the long term.

## 5. Conclusions

In conclusion, our small but informative proof-of-concept prospective study showed no significant differences in the ventricular remodeling assessed by CMR between no OSA and OSA patients of different severity at six months after the AMI. These results raise questions about the target population and the optimal timing for evaluating sleep and apnea after the AMI and highlight the complexity of the relationships between OSA and post-AMI morbi-mortality.

## Figures and Tables

**Figure 1 jcm-11-06341-f001:**
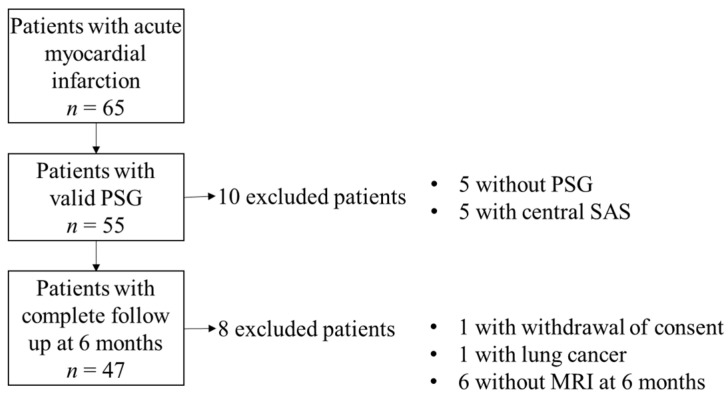
Flow Chart.

**Figure 2 jcm-11-06341-f002:**
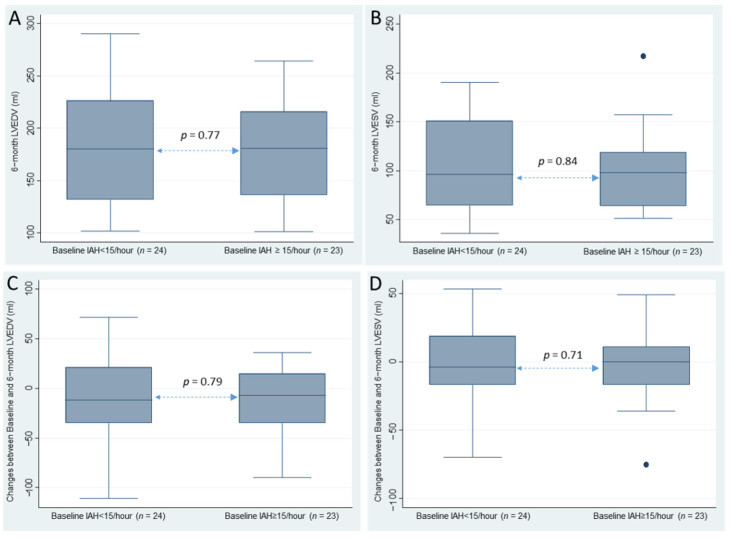
Differences between patients with an AHI < 15/h and patients with an AHI ≥ 15/h for: (**A**) six-month left ventricular end-diastolic volume (LVEDV); (**B**) left ventricular end-systolic volume (LVESV); (**C**) changes between baseline and six months in LVEDV; and (**D**) LVESV.

**Figure 3 jcm-11-06341-f003:**
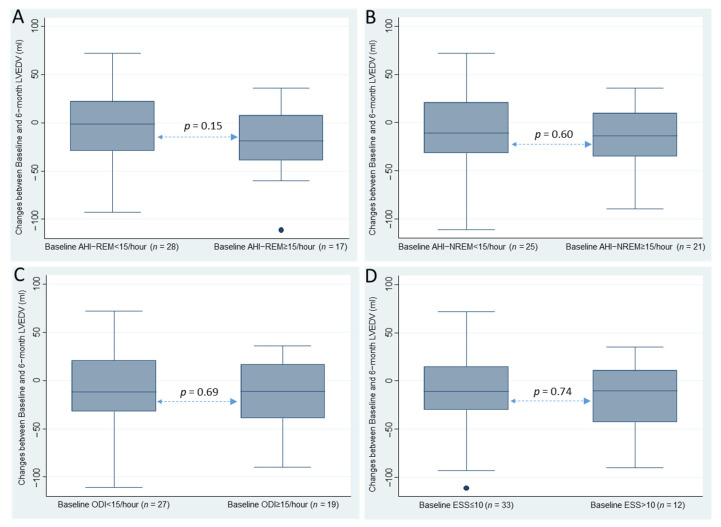
Changes between baseline and six months in LVEDV between patients with: (**A**) AHI-REM < 15/h and ≥15/h; (**B**) AHI-NREM < 15/h and ≥15/h; (**C**) oxygen desaturation index (ODI) < 15/h and ≥15/h; and (**D**) Epworth sleepiness scale (ESS) ≤ 10 and >10.

**Table 1 jcm-11-06341-t001:** Baseline demographic, clinical, biological characteristics based on the severity of the apnea–hypopnea index.

	Apnea–Hypopnea Index		
	<15/h *n* = 24	≥15/h *n* = 23	
Variables			*p*
Female, Gender, *n* (%)	5 (20.83)	2 (8.70)	0.42
Age, years ^(1)^	54.31 (33.60; 70.38)	60.65 (42.22; 74.41)	0.09
BMI, kg/m^2^ ^(1)^	24.58 (18.52; 32.65)	26.18 (20.76; 33.14)	0.14
Systolic BP, mmHg ^(1)^	111.50 (79.00; 163.00)	121.00 (90.00; 162.00)	0.20
Diastolic BP, mmHg ^(1)^	72.00 (28.50; 100.00)	69.00 (10.10; 99.00)	0.38
Diabetes, Yes, *n* (%)	3 (12.50)	1 (4.55)	0.61
Hypertension, Yes, *n* (%)	8 (33.33)	9 (39.13)	0.68
Hypercholesterolemia, Yes, *n* (%)	5 (21.74)	11 (50.00)	0.05
Current smoker, Yes, *n* (%)	12 (50.00)	10 (45.45)	0.76
Family history of CA, *n* (%)	10 (41.67)	8 (36.36)	0.71
TIMI-flow before PCI			0.05
Grade 0	16 (84.21)	9 (64.29)	
Grade 1	0 (0.00)	4 (28.57)	
Grade 2	3 (15.79)	1 (7.14)	
TIMI-flow after PCI			0.46
Grade 0	0 (0.00)	1 (5.26)	
Grade 3	22 (100.00)	18 (94.74)	
hs-CRP, mg/L ^(1)^	24.40 (0.30; 106.40)	11.70 (1.30; 128.60)	0.01
hs-cTnT, ng/L ^(1)^	2870.50 (10.70; 17,459.00)	3594.00 (6.10; 14,445.00)	0.64
CPK, U/L ^(1)^	782.00 (126.00; 9272.00)	21; 414.00 (122.00; 5182.00)	0.41
ACE inhibitor/ARB intake, Yes, *n* (%)	22 (91.67)	21 (91.30)	0.99
β-blocker intake, Yes, *n* (%)	22 (91.67)	19 (82.61)	0.42
Statins intake, Yes, *n* (%)	22 (91.67)	21 (91.30)	0.99
ESS ^(1)^	6.00 (0.00; 15.00)	6.00 (0.00; 19.00)	0.63
ISI score ^(1)^	8.00 (0.00; 23.00)	7.00 (0.00; 23.00)	0.26
BDI-II total score ^(1)^	9.50 (0.00; 39.00)	4.00 (0.00; 22.00)	0.08
EQ5D-VAS ^(1)^	50.00 (30.00; 90.00)	65.00 (40.00; 90.00)	0.02

^(1)^ continuous variables were expressed as a number; median (minimum value, maximum value). Abbreviations: ACE: angiotensin-converting enzyme; ARB: angiotensin receptor blockers; BMI = body mass index; BDI-II = Beck depression inventory-II; CPK: creatine phosphokinase; CRP: C-reactive protein; EQ5D = European quality of life five-dimensions questionnaire, ESS = Epworth sleepiness scale; hs-CRP: high sensitivity C-reactive protein; hs-cTnT: high-sensitivity cardiac troponin T; ISI: insomnia severity index; PCI: percutaneous coronary intervention; TIMI: thrombolysis in myocardial infarction; VAS = visual analog scale.

**Table 2 jcm-11-06341-t002:** Baseline polysomnographic characteristics based on the severity of apnea–hypopnea index.

	Apnea–Hypopnea Index		
	<15/h *n* = 24	≥15/h *n* = 23	
Variables	Median (Min-Max)	Median (Min-Max)	*p*
Hypopnea index	5.80 (0.63; 14.70)	16.68 (3.87; 38.19)	<0.0001
Obstructive sleep apnea index	0.15 (0.00; 1.23)	3.54 (0.00; 15.39)	<0.0001
Central sleep apnea index	0.14 (0.00; 4.64)	2.71 (0.00; 17.22)	<0.0001
Apnea–hypopnea index in REM	10.65 (0.00; 46.29)	19.38 (1.37; 59.98)	0.02
Apnea–hypopnea index in NREM	4.06 (0.00; 11.20)	31.39 (13.07; 58.47)	<0.0001
Oxygen desaturation index	5.56 (0.33; 16.90)	21.18 (4.09; 51.21)	<0.0001
SpO_2_, %	95.00 (91.00; 98.00)	95.00 (87.00; 97.00)	0.65
Microarousal index, per hour	11.29 (4.03; 84.00)	22.26 (4.66; 46.87)	0.0001
N3 (%)	19.27 (4.90; 35.84)	15.38 (0.00; 35.16)	0.23
Total sleep time (minutes)	379.00 (229.00; 517.00)	373.00 (227.00; 640.00)	0.75
Sleep efficiency (%)	79.03 (40.63; 95.07)	76.44 (45.91; 94.44)	0.92

Abbreviations: NREM = nonrapid eye movement sleep; REM = rapid eye movement sleep; N3 = non-REM sleep stage 3; SpO_2_: peripheral oxygen saturation.

**Table 3 jcm-11-06341-t003:** Changes in cardiac remodeling between baseline and a six-month follow-up in patients with baseline apnea hypopnea index < 15/h and patients with baseline apnea hypopnea index ≥ 15/h.

	Among Patients with Baseline AHI < 15/h *n* = 24			Among Patients with Baseline AHI ≥ 15/h *n* = 23				
	At Baseline	At Six-Month Follow-Up		At Baseline	At Six-Month Follow-Up		Between-Group Difference Δ Baseline—6 Months	
	Median (IQR)	Median (IQR)	*p*	Median (IQR)	Median (IQR)	*p*	Median (95% CI) ^(1)^	*p*
LV end-diastolic volume (ED) (mL)	183.50 (50.50)	180.50 (95.00)	0.22	155.00 (56.00)	181.00 (80.00)	0.11	3.50 (18.00; 24.00)	0.79
LV end-systolic volume (mL)	110.00 (55.00)	96.00 (86.50)	0.92	95.00 (41.00)	98.00 (55.00)	0.42	3.50 (−14.00; 19.00)	0.71
LVEF (%)	42.00 (12.00)	45.00 (18.00)	0.09	42.00 (8.00)	46.00 (14.00)	0.25	−1.00 (−6.00; 4.00)	0.66
LV ejection volume (mL/m^2^)	39.00 (8.00)	44.60 (13.85)	0.05	34.20 (7.70)	42.00 (14.00)	0.13	−2.00 (−7.80; 5.00)	0.60
LV cardiac output (L/min)	4.68 (1.16)	4.66 (1.38)	0.93	4.70 (1.09)	4.59 (1.29)	0.77	−0.11 (−0.87; 0.55)	0.68
LV cardiac index (L/min/m^2^)	2.52 (0.43)	2.35 (0.62)	0.91	2.42 (0.64)	2.47 (0.65)	0.55	−0.04 (−0.41; 0.32)	0.87
RV end-diastolic volume index (mL)	120.50 (43.50)	128.50 (35.00)	0.48	115.00 (31.00)	129.00 (57.00)	0.31	3.00 (−15.00; 22.00)	0.68
RV end-systolic volume (mL)	49.50 (26.00)	54.50 (18.50)	0.65	52.00 (26.00)	54.00 (37.00)	0.93	−0.50 (−11.00; 10.00)	0.92
RVEF (%)	58.50 (6.50)	59.00 (10.50)	0.60	56.00 (8.00)	57.00 (14.00)	0.20	3.00 (−2.00; 8.00)	0.22

Abbreviations: LV = left ventricular; LVEF = left ventricular ejection fraction; RV: right ventricular; RVEF: right ventricular ejection fraction. ^(1)^ Medians of the differences between baseline and six months between patients with an AHI < 15/h and patients with an AHI ≥ 15/h.

## Data Availability

Data can be provided upon reasonable request.

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
