# Peer review of "Impact of Obstructive Sleep Apnea Syndrome on Ventricular Remodeling after Acute Myocardial Infarction: A Proof-of-Concept Study"

_jcm, 2022, doi:10.3390/jcm11216341_

Round 1

Reviewer 1 Report (Previous Reviewer 2)

All comments have been adequately addressed.

Author Response

We thank the reviewer once again for these constructive comments.

Reviewer 2 Report (New Reviewer)

Sleep apnoea syndrom is an independent risk factor for cardiovascular disease. There are evidence of association

between obstructive sleep apnoea syndrome and cardiovascular diseases; systemic arterial hypertension, ischaemic heart disease, atrial fibrillation, stroke, heart failure, and cardiac sudden death.

The authors try to demonstrate the effect of OSAS on heart remodeling in patients after myocardial infarction.

The study is properly planned, the methodology allows for obtaining objective results. The statistical methods are correctly selected. The main limitation of the study is the small number of study groups, which the authors themselves emphasize. However, to be aware of the limitations of the process, appropriate balanced conclusions are drawn.

Minor revision:

Table 3 - The penultimate column, "Between group difference", is the difference in baseline or 6 months ?.

Author Response

We thank the reviewer for these positive comments.

The penultimate column, "Between group difference", includes the medians of the differences between baseline and 6 months between patients with AHI<15/hour and patients with AHI≥15/hour. We have completed the table 3 with this important clarification.

Kind regards.

Pr Dauvilliers, Pr Roubille and Dr François Bughin

This manuscript is a resubmission of an earlier submission. The following is a list of the peer review reports and author responses from that submission.

Round 1

Reviewer 1 Report

Dear Sir/Madam,

I had the opportunity to act as a reviewer on the recent submission by Bughin et al. to the Journal of Clinical Medicine.

The authors present original research studying the effect of obstructive sleep apnea syndrome on the ventricular remodeling after acute myocardial infarction. They found no association between obstructive sleep apnea and ventricular remodeling after acute myocardial infarction.

The manuscript is well structured. However, the biggest issue is, as the authors in the limitations section already describe, the sample size of 47 patients, who completed the follow-up and were included in the final analysis. The number is lower than the planned sample size of 87 patients, which impedes drawing an adequate conclusion. I recommend including more patients in the study and further analysis.  

Best regards,

Author Response

We agree with the reviewer. The main limitation of our study is the sample size with risk of type II error, with low recruitment of 47 patients (65% of the planned sample size), with 10% lost during follow-up, with wide range in BMI, with underrepresentation of females, and the monocentric study design. The low sample size can be explained by the absence of routine CMR in patients diagnosed with AMI, the inclusion of severe AMI patients (baseline CMR delayed enhancement score ≥5 segments), the inability to recruit patients during the Covid-19 pandemic, and by difficulties for patients to come back for a CMR reevaluation when living far away from our tertiary center. However, achieving the expected sample size would have shown no significant effect of OSA severity on the post-AMI ventricular remodeling. A post hoc power analysis showed that 1 032 patients would be required to reach a power of 0.8 with a significant level of 0.05 considering a mean LVEDV=189 mL for the 7 patients with an IAH<5/hour and a mean LVEDV=180.38 mL for the 40 patients with an IAH≥5/hour and a standard deviation at 51.54 for the 47 patients.  Such recruitment is beyond the scope of the present study.

Based on this power analysis, we decided to not pursue recruitment for this costly and time-consuming study. These points have been included in the discussion section.

Reviewer 2 Report

I had the pleasure of reviewing “Impact of Obstructive Sleep Apnea Syndrome on the Ventricular Remodeling AfterAcute Myocardial Infarction” by Francois Bughin et al. The authors performed a prospective study in patients with acute myocardial infarction, including baseline and 6-month cardiac magnetic resonance imaging. Furthermore, they performed polysomnography and associated the presence of obstructive sleep apnea syndrome (OSAS) with ventricular remodeling. The study unfortunately was negative.

First, I want to congratulate the authors for performing such an interesting prospective study, which has for sure been quite difficult. Despite the negative results, in my opinion it is still important to publish these data, also to further support existing data (such as the ISAAC study, which was larger). In general, the manuscript is well written with adequate statistics and the conclusions are backed up by the results.

Due to the separation into 4 groups based on the Apnea-Hypopnea index and then additionally into 2 groups, the study is quite complicated to read. I would recommend to stay with the threshold of 15/h without any further segregation.

In total, the manuscript is well-written with adequate statistics, conclusions and references.

Minor comments:

-        Abstract: The proportion of patients with OSAS should be presented.

-        The proportion of STEMI should be mentioned in the text.

-        The tables are currently very difficult to read and have many columns. Furthermore, in metric parameters I would skip the count (n=), except for parameters with missing values.

-        Figure 2 also contains very small fonts (please increase size)

-        Table 3: This table could be improved by removing the columns “at baseline” and “at 6-months follow-up” (2x each)

-        A new section “strengths and limitations” with a separate title may be added.

-        Conclusions: In comparison to ISAAC the study may be small, but it was still prospective and included cMRI. I would suggest to add these positive methodological details to the conclusion.

Author Response

Dear Reviewer,

We are very pleased to submit the responses to the comments for our manuscript entitled “Impact of Obstructive Sleep Apnea Syndrome on the Ventricular Remodeling After Acute Myocardial Infarction”.

The authors wish to thank the reviewers for their constructive comments.

Best regards, I had the pleasure of reviewing “Impact of Obstructive Sleep Apnea Syndrome on the Ventricular Remodeling After Acute Myocardial Infarction” by Francois Bughin et al. The authors performed a prospective study in patients with acute myocardial infarction, including baseline and 6-month cardiac magnetic resonance imaging. Furthermore, they performed polysomnography and associated the presence of obstructive sleep apnea syndrome (OSAS) with ventricular remodeling. The study unfortunately was negative.

First, I want to congratulate the authors for performing such an interesting prospective study, which has for sure been quite difficult. Despite the negative results, in my opinion it is still important to publish these data, also to further support existing data (such as the ISAAC study, which was larger). In general, the manuscript is well written with adequate statistics and the conclusions are backed up by the results.

Due to the separation into 4 groups based on the Apnea-Hypopnea index and then additionally into 2 groups, the study is quite complicated to read. I would recommend to stay with the threshold of 15/h without any further segregation

We changed the abstract, the results and the tables accordingly.

In total, the manuscript is well-written with adequate statistics, conclusions and references.

Minor comments:

-        Abstract: The proportion of patients with OSAS should be presented.

We added this result in the abstract “49% of patients had moderate or severe OSA with an AHI ≥15/h. No differences were observed between these patients and those with AHI <15/hour for left ventricular end-diastolic and end-systolic volumes at 6 months”

-        The proportion of STEMI should be mentioned in the text.

All patients had ST-elevation myocardial infarction (STEMI).  We added this in the results.

-        The tables are currently very difficult to read and have many columns. Furthermore, in metric parameters I would skip the count (n=), except for parameters with missing values.

We simplified the tables accordingly. Given the small amount of missing data, we removed all the n from the tables.

        Figure 2 also contains very small fonts (please increase size)

Indeed, we have increased the size of figures 2 and 3 to facilitate reading.

-        Table 3: This table could be improved by removing the columns “at baseline” and “at 6-months follow-up” (2x each)

We have simplified table 3 for better readability.

-        A new section “strengths and limitations” with a separate title may be added.

We have added this section as proposed.

-        Conclusions: In comparison to ISAAC the study may be small, but it was still prospective and included cMRI. I would suggest to add these positive methodological details to the conclusion.

      We added these points in the conclusion.

Round 2

Reviewer 1 Report

-